

# Characterization of Local Wind Profiles: A Random Forest Approach for Enhanced Wind Profile Extrapolation

Farkhondeh (Hanie) Rouholahnejad[1] and Julia Gottschall[1]

[1]Fraunhofer Institute for Wind Energy Systems IWES, Am Seedeich 45, 27572 Bremerhaven, Germany

**Correspondence:** Farkhondeh Rouholahnejad (farkhondeh.rouholahnejad@iwes.fraunhofer.de)

**Abstract.** Accurate wind speed determination at the height of the rotor swept area is critical for resource assessments. ERA5 data combined with short-term measurements through the "Measure, Correlate, Predict" (MCP) method is commonly used for offshore applications in this context. However, ERA5 poses limitations in capturing site-specific wind speed variability due to its low resolution. To address this, we developed random forest models extending near-surface wind speed up to 200 m, focusing on the Dutch part of the North Sea. Our results show that the random forest model trained on site-specific wind profiles outperforms the MCP-corrected ERA5 wind profiles in accuracy, bias, and correlation. In absence of rotor-height measurements, a model trained within a 200 km region handles vertical extension effectively, albeit with increased bias. Our regionally trained random forest model exhibits superior accuracy in capturing wind speed variations and local effects, with an average deviation below 5% compared to corrected ERA5 with a 20% deviation from measurements. The random forest model adeptly captures the inertial subrange of the power spectrum where ERA5 shows degradation. Our study highlights the potential enhancement in wind resource assessment by means of machine learning methods, specifically random forest. Future research may explore extending the random forest methodology for higher heights, benefiting new generation of offshore wind turbines, and investigating cluster wakes in the North Sea through a multinational network of floating lidars, contingent on data availability.

## 1 Introduction

Accurate wind speed knowledge across the entire turbine swept area is paramount for the wind energy industry, specifically for site assessment and energy yield calculations (Rohrig et al., 2019). Direct wind profile measurements remain the gold standard, with remote sensing devices like lidars, especially floating lidars, gaining popularity offshore for their ability to reach heights beyond traditional meteorological masts and reduce costs (Gottschall et al., 2017). However, like meteorological masts, they provide wind profiles as point measurements, corresponding to specific locations in space without comprehensive spatial coverage.

In contrast, meso-scale models and global reanalysis datasets offer extensive horizontal coverage but are hindered by spatial and temporal resolutions and the associated errors. Especially in offshore locations, where measurements are often proprietary and scarce, these models remain insufficiently validated (Hahmann et al., 2015). Moreover, due to their large spatial and temporal resolutions, these models tend to smooth out wind speed fluctuations similar to a low-pass filter, thereby inaccurately





predicting wind speed variability—an essential input for turbine design (Dörenkämper et al., 2020; IEC, 2019). As a result, wind farm developers commonly adopt a hybrid approach, combining the strengths of both direct measurements and modeling (Carta et al., 2013). This involves conducting measurement campaigns spanning months to a year (Strack et al., 2010), utilizing this data to refine and correct modeled information and extend it over the planned wind turbines' lifespan. The "Measure,

Correlate, Predict" (MCP) method is often employed in this context to localize modeled data at the desired site (Strack et al., 2010). Given that the North Sea exhibits one of the highest levels of wind speed fluctuations in Europe (Bett et al., 2013), there is a need for novel methods, more accurate than the conventional MCP method, to provide more localized wind speed estimates and lower investment risks (Lee and Fields, 2021).

Recent studies have indicated the potential of machine learning (ML) methods, particularly random forests, in extrapolating

wind speed to the height of the rotor swept area. Mohandes and Rehman (2018) employed deep neural networks (DNN) to extrapolate lidar measurements in flat terrain. Taking this methodology a step further, Vassallo et al. (2020) analyzed the sensitivity of DNN to input features in complex terrains, achieving up to 65% and 53% accuracy improvement compared to log-law and power-law predictions, respectively. Notably, these studies trained and tested machine learning models at the same location. Bodini and Optis (2020b) argued that this approach is neither fair nor practical. It is unfair because conventional

models as e.g., log law and power law or meso-scale models, do not see the wind speeds at the heights of prediction, and it is impractical because there is no need to predict wind speed where the wind profiles are already known, i.e., at the training location. Hence, they introduced the round robin validation method to the literature, defined as testing the ML model at a location distant from the training. They implemented this validation method on data collected at four onshore locations in a 100 km wide region in the USA. They showed that the random forest-predicted wind profiles fed with near-surface measurements

and wind speed at 65 m can improve the predictions of log and power law by 25%, reducing to 17% when round robin is accounted for. Random forest has also been used to vertically extrapolate wind speed offshore. Optis et al. (2021) utilized two 83 km apart floating lidars in the North Atlantic of the U.S. offshore area to develop random forests, extending near-surface speed up to 200 m and evaluating the results on a climatological level. The round robin approach increased the unbiased root mean squared error (RMSE) by 6-9% but still outperformed the Weather Research and Forecasting model (WRF) in all

stability conditions, seasons, and times of day. In a similar vein, Rouholahnejad et al. (2023) adopted the round robin approach to validate random forest models, extending wind speed up to 300 m. They utilized fixed lidars on three offshore platforms in a 300 km wide region in the North Sea. Their round robin approach resulted in a 14% improvement in the mean absolute error for the region-optimized WRF model. Hatfield et al. (2023) also considered the round robin approach to extrapolate satellite wind speed retrievals from 10 m to 100 m using random forest models in the North and Baltic Sea, achieving a 35% improvement

in RMSE compared to NEWA (New European Wind Atlas), albeit facing the challenge of low data availability from satellites (defined by 2-4 overpasses per day only). The evaluation of the performance of the random forest algorithm is not only limited to the accuracy of the predicted time series. Bodini and Optis (2020a) and Hallgren et al. (2023) showed that random forest is also able to capture low-level jets, an important phenomenon for wind energy applications. These studies have showcased the potential of random forests in accurately extending the wind profile in space; however, they do not address its ability to

capture wind speed variability. To bridge this knowledge gap, this work will address the following research questions: Firstly,



how accurately can random forest predict wind speed variability and structures of different frequencies? And secondly, how does random forest compare with the currently used MCP method in the resource assessment context?

To address these questions, we developed a random forest-based methodology using measured wind profiles in the North Sea, aiming to overcome the issues associated with the low temporal and spatial resolution of ERA5 and provide more localized wind speed predictions. In Section 2, we introduce the floating lidar settings near the Netherlands coast, proposing two random forest-based methodologies using near-surface measurements to predict the wind profile up to 200 m. Our validation process involves comparing the ML model with MCP-corrected ERA5 profiles, elaborated upon in this section. Section 3 presents the validation results and explores horizontal extrapolation via a round-robin to test the ML model's robustness and generality in the region. These two validation approaches contribute to our understanding of random forest's potential in improving conventional correction methods for site resource assessment. The results are discussed in Section 4. Finally, Section 5 summarizes and concludes this study, followed by insights into the future of this research field.

## 2    Material and methods

In this section, we describe the collected observations and present a methodology to develop a model for extending wind profiles at four offshore sites located in the Dutch part of the North Sea. The model's predictions are benchmarked against wind profiles obtained from ERA5 pressure levels, and both are validated against lidar measurements at site.

### 2.1    Observational data

A reliable dataset is key to train and validate a data-driven model. In this study, we utilize measurements obtained by the SEAWATCH Wind Lidar Buoys (SWLB) deployed by Fugro Norway AS within the wind farm zones of Dutch waters (DNV; Netherland Enterprise Agency). The SWLB, also referred to as Floating Lidar System (FLS), collected wind and wave data at four specific locations: Hollandse Kust (west, noord, zuid) and Ten noorden van de Waddeneilanden, in this study referred to as HKW, HKN, HKZ, and TNW respectively (see Fig. 1).

Each SWLB is equipped with a sonic anemometer, which measures wind speed and direction at 4 m above mean sea level (aMSL). Additionally, a ZX lidar system (previously ZephIR) is mounted on the buoy. This continuous wave (CW) lidar system measures wind speed and direction at 10 ranges between 30 and 250 m aMSL. For the HKW and TNW sites, wind speed is measured up to 250 meters, while for HKN and HKZ, measurements extend up to 200 meters. To maintain uniformity in our analysis, we have opted to standardize the highest wind speed height to 200 meters across all sites, ensuring consistency in our study.

In addition to wind speed and direction, the Vaisala air pressure, temperature, and humidity sensors, installed on the FLS, measure the atmospheric conditions. The water temperature is measured at 1 m below the surface, which we take as the sea surface temperature (SST). All variables are available at a 10-minute resolution and are time-stamped at the end of each averaging period. The measured variables used in this study, along with their associated heights, are summarized in Table 1. We used the lidar wind direction at 100 m to excluded the wind sectors, where wind farms were operating close-by in order





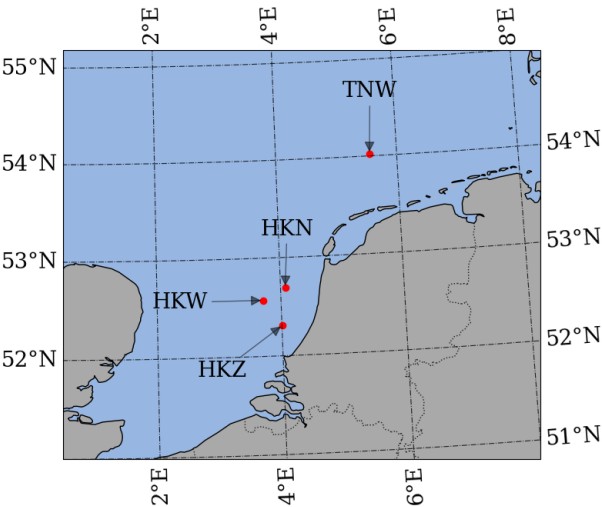

**Figure 1.** Map showing the locations of the sites. HKW, HKN, HKZ and TNW are acronymes for Holland Kuste West, Noord, Zuid, and Ten noorden van de Waddeneilanden

**Table 1.** Variables measured by the Floating Lidar Systems (FLS) in this study, along with their associated heights. This information applies uniformly to all FLS units used, given their consistent and similar structure. 'aMSL' signifies above mean sea level.

| Variable | Device | Height [m aMSL] |
|---|---|---|
| Pressure | Vaisala PTB330A | 0.5 |
| Wind speed and direction | ZephIR 300S / ZX 300 | 30, 40, 60, 80, 100, 120, 140, 160, 180, 200 |
| Wind speed and direction | Gill Windsonic M acoustic wind sensor | 4 |
| Temperature and humidity | Vaisala HMP155 | 4 |
| SST | Nortek Aquadopp 600kHz current profiler | -1 |

to remove their wakes. Detailed information regarding the location, duration, and the data availability is listed in Table 2. The data availability corresponds to the concurrent lidar profiles and the met station measurements after applying the wind sector filter. It is important to note that between the two (at HKN, HKZ, TNW) or three (at HKW) deployed FLS, the one with the highest data availability is chosen for this study and no data gap filling incorporating neighbouring buoys or other data sources is implemented. However, the data gap of sea surface temperature on buoy A at HKN starting from 12.01.2019 for one and a half months is filled with the one measured by buoy B. We assumed that the sea surface temperature is fairly constant over a distance of 2 km.



**Table 2.** Summary of campaign details: location, duration, and data availability for each FLS. The Data availability corresponds to the concurrent profiles.

| Site | Buoy | Campaign period | Lat, Lon | Coastal dir. & dist. | Direction filter (not valid) | Data availability |
|------|------|-----------------|----------|----------------------|------------------------------|-------------------|
| HKW | A | 05.02.2019 - 11.02.2021 | 52.57°, 3.72° | East, 53 km | – | 83 % |
| HKN | A | 10.04.2017 - 10.04.2019 | 52.69°, 4.08° | East, 18.5 km | 100-200° | 60 % |
| HKZ | B | 05.06.2016 - 05.06.2018 | 52.31°, 4.01° | East, 18 km | 20-70° | 80 % |
| TNW | A | 19.06.2019 - 20.06.2021 | 54.02°, 5.56° | South, 56 km | 60-120° | 64 % |

## 2.2 Reanalysis data

The proposed methodology is benchmarked against ERA5, the state-of-the-art fifth generation reanalysis dataset, developed by ECMWF. ERA5 combines historical measured data with numerical models using the IFS Cycle 41r2 data assimilation model in 12 hour windows to provide hourly atmospheric variables since 1940 (Hersbach et al., 2020; ECMWF). It offers a spatial resolution of $0.25° \times 0.25°$ covering the entire globe, hence making it a powerful tool to be used for wind resource assessment, energy yield calculations, or climate change studies. In the specific region of this study, the spatial resolution corresponds to $28 \times 17$ km (latitude and longitude).

We extracted ERA5 pressure level data available at the closest grid point to the buoy. These profiles are then interpolated to obtain the horizontal wind speed at the desired heights by means of fitting of a monotone cubic function to the wind profile. The ERA5 wind profiles are corrected based on the method elaborated in section 2.4.

## 2.3 Random forest based models

In this study, we introduce two models based on the random forest algorithm: the random forest regressor (Breiman, 2001) and quantile regression forest (Meinshausen, 2006). These models are employed to extend wind speed measurements collected by the sonic anemometers. Our methodology involves randomly selecting a continuous 15% of the data each month for testing, while utilizing the remaining data for training the models. It is important to note that fully random splitting assumes that the dataset is representative, with observations being independent and identically distributed—conditions that do not hold in our case. Therefore, our approach prevents the introduction of artificial correlations between the testing and training datasets, ensuring the preservation of the underlying seasonality, while still allowing for accurate model evaluation. To tune the hyperparameters, we chose a 15% continuous subset within the training period and optimized for the RMSE of this subset. Table 3 provides the hyperparameters for both random forest regressor and quantile regression forest models at all sites.

We conducted both same-site and round-robin validations for each model. In "round-robin validation", the model is applied to locations where it was not initially trained. This approach extends wind profiles spatially. Meanwhile, "same-site validation" extends the near surface wind speed vertically (Bodini and Optis, 2020b).



The random forest models' outputs and the measured data are down-sampled to match the temporal resolution of ERA5 and stamped in the middle of the period.

### 2.3.1 Random forest regressor - RF

The random forest algorithm consists of multiple regression decision trees, each trained on bootstrapped data. This ensemble approach enhances the robustness of the model. Decision trees, which are a type of supervised learning model, make decisions by recursively splitting the data at each node based on the best feature. The best feature is determined by its ability to minimize the mean squared error, when used to split the data into subsets. The splitting process continues until a specific criterion is met, at which point a node becomes a leaf. Each leaf in the tree can contain multiple observations (see Fig. 2).

In a random forest model, the prediction of a regression tree is the conditional mean of the observations within the corresponding leaf that the test sample falls into when traversing down the tree. By averaging the predictions of multiple trees, we obtain the prediction of the random forest model. We used the RandomForestRegressor class from sklearn (Pedregosa et al., 2011).

### 2.3.2 Quantile regression forest - QRF

In contrast to random forest, where only the mean of the observations in the leaf nodes is retained, a quantile regressor forest preserves the entire data distribution. As a result, the prediction of each tree in a quantile regressor forest corresponds to the desired conditional quantile of the observations at each node. This is particularly useful, as it provides information regarding the confidence interval and allows for the use of the median of the observations to predict the profile, rather than relying on the mean as in the case of random forest. We used the RandomForestQuantileRegressor class from the sklearn_quantile package (Roebroek, 2022), which is programmed based on an algorithm proposed by Meinshausen (2006). The hyperparameters used to create the quantile regression forests are listed in Table 3. We used the minimum sample leaf to ensure that the median is representative by having sufficient number of samples at each leaf.

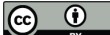

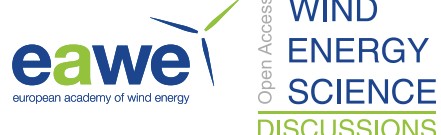

a)

b)

c)

**Figure 2.** Overview of the Random Forest Regressor (RF) and Quantile Regression Forest (QRF) – (a) depicts the structure of an example tree, with the green path indicating the route taken by a testing sample; (b) illustrates the sample distribution of the leaf node where the testing sample landed, highlighting distinctions between RF and QRF predictions; (c) presents the actual predictions of RF and QRF models trained at HKW, along with the confidence interval derived from the QRF model.





**Table 3.** Hyperparameters used in the random forest and quantile regression forest models.

| Site | Hyperparameter | RF | QRF |
|------|----------------|----|----|
|     | n_estimator | 100 | 100 |
| HKW | max_feature | 7 | 8 |
|     | min_sample_leaf | 40 | 40 |
|     | n_estimator | 300 | 100 |
| HKN | max_feature | 7 | 7 |
|     | min_sample_leaf | 50 | 60 |
|     | n_estimator | 300 | 100 |
| HKZ | max_feature | 8 | 8 |
|     | min_sample_leaf | 40 | 50 |
|     | n_estimator | 300 | 100 |
| TNW | max_feature | 7 | 7 |
|     | min_sample_leaf | 20 | 30 |

## 2.4 Measure-Correlate-Predict

145 The "Measure, Correlate, Predict" (MCP) method is an established technique used in wind resource assessment to estimate long-term wind characteristics at a specific location (Carta et al., 2013; Rogers et al., 2005). Due to the high cost of measurement campaigns, especially offshore, wind data is typically collected for a shorter period (typically one year) and then correlated with reference data from nearby locations or reanalysis/meso-scale data. This correlation helps correct the reference data in the absence of in-situ measurements, providing a more accurate representation of long-term wind characteristics at the 150 specific location.

In this study, we used the correlation between the ERA5 and measured wind speeds during training to correct the ERA5 testing subset for the same-site approach. For each height individually, we employed a two-parameter linear regression model (slope and intercept) by minimizing the least squared error with respect to the training subset. Subsequently, we utilized the derived slope and intercept values to adjust the testing subset accordingly. This approach ensures a fairer comparison between 155 ERA5 and the random forest model since both models have and use knowledge about the training subset.

## 2.5 Statistical parameters

To assess the accuracy of wind speed predictions, we employed standard error metrics. Additionally, we computed ramp rates to gauge wind speed variability and analyzed the power spectrum to gain insights into the underlying structures present in the wind data.



### 2.5.1 Error metrics

We used the root mean squared error (RMSE), mean absolute error (MAE), Bias and the coefficient of correlation $R^2$ as defined below to estimate the overall performance of the predicted time series.

$$\text{RMSE} = \sqrt{\frac{1}{N}\Sigma(U^t_{\text{model}} - U^t_{\text{obs}})^2}, \tag{1}$$

$$\text{MAE} = \frac{1}{N}\Sigma|U^t_{\text{model}} - U^t_{\text{obs}}|, \tag{2}$$

$$\text{Bias} = \frac{1}{N}\Sigma(U^t_{\text{model}} - U^t_{\text{obs}}), \tag{3}$$

$$\text{R}^2 = 1 - \frac{\Sigma(U^t_{\text{model}} - U^t_{\text{obs}})^2}{\Sigma(U^t_{\text{model}} - \overline{U_{\text{obs}}})^2}, \tag{4}$$

where $U^t_{\text{model}}$ and $U^t_{\text{obs}}$ are the predicted and measured wind speed at time stamp $t$, and $\overline{U_{\text{obs}}}$ is the mean observed wind speed over time.

### 2.5.2 Wind speed ramp rate

To evaluate the ability of the models to capture the variability of the site, we calculated the ramp rate as the change of the wind speed in a one hour period (Milan et al., 2014):

$$\text{Ramp rate} = U^{t_i} - U^{t_{i-1}}. \tag{5}$$

The temporal resolution of the concurrent data set is one hour, hence we present the hourly ramp rates. But to gain a general understanding of the hourly fluctuations, we calculated the mean absolute hourly ramp rate as:

$$\mu_{\text{ramp rate}} = \frac{1}{N}\Sigma|U^{t_i} - U^{t_{i-1}}|. \tag{6}$$

### 2.5.3 Power spectral density

The power spectral density (S) was computed using the Fourier transform ($\mathcal{F}$) of the detrended horizontal wind speed as follows:

$$S = \frac{2}{N} \cdot \lim_{T \to \infty} \Sigma|\mathcal{F}(U^t - \overline{U})|^2, \tag{7}$$

where $T$ is the period, and $N$ is the number of samples. The defined $S$ in Eq. 7 is for positive frequencies, representing the one-sided spectrum. To smooth the spectrum, hamming windows of 30 days without overlap were applied.

In this study, we present the measured and modeled $S$ for the entire data period, approximately 2 years for each site. This approach allows for a more continuous time series and helps avoid filling gaps. It is important to note that we did not apply the RF model if it had seen the 15% training subset to ensure unbiased spectral comparisons.





## 2.6 Bulk Richardson number

Atmospheric stability refers to the condition of the atmosphere in terms of its tendency to resist vertical motion or mixing of air masses. The bulk Richardson number ($R_B$) relates the buoyancy force to the shear force and therefore can give insights on the stability condition. We calculated $R_B$ described by Eq (8), where $\theta_v$ and $g$ are the virtual potential temperature and the acceleration due to gravity, respectively. $z$ is the height (here 4 m) and $U_z$ represents the horizontal velocity at this height (Stull, 1988). Here we took the parameters at sea surface and at height $z$, and assumed no slip condition at sea surface (zero velocity). For a more detailed derivation, we refer to the appendix A.

$$R_B = \frac{g \Delta \theta_v \Delta z}{\theta_v U_z^2}. \tag{8}$$

Having the bulk Richardson number, the stability parameter ($\zeta$) can be estimated as follows (Grachev and Fairall, 1997):

$$\zeta = \begin{cases} \frac{C_1 R_B}{1 - C_2 R_B} & \text{for } R_B > 0 \\ C_1 R_B & \text{for } R_B \leq 0, \end{cases} \tag{9}$$

where $C_1$ and $C_2$ are 10 and 5 respectively. Finally the Monin-Obukhov length ($L$) can be calculated as:

$$L = \frac{\frac{1}{2} z}{\zeta}. \tag{10}$$

$L$ was utilized in the post-processing phase to examine the impact of stability on the predictions. The samples are categorized as stable if $0 < L < 1000$, and unstable if $-1000 < L < 0$.

## 3 Results

In our investigation, we implemented the proposed methodology by training four distinct random forest models across four geographically diverse sites. This section starts with a detailed discussion of the crucial features embedded in these models. Subsequently, we present and compare the predicted wind profiles with the corresponding ERA5 profiles. To evaluate model performance, lidar profiles serve as ground truth, and an array of performance metrics is computed. Furthermore, a meticulous analysis of error characteristics provides profound insights into the potential of the applied machine learning algorithms, in terms of wind speed variability and power spectrum.

### 3.1 Features

The normalized importance of each input for the random forest algorithm, as depicted in Figure 3 (left), is determined by its role in reducing squared error. Consistent with established literature (Optis et al., 2021; Bodini and Optis, 2020b), our study reaffirms that near-surface wind speed stands out as the most crucial input. Our analysis further underscores the significance of the air-sea temperature difference as a proxy for stability, aligning with prior research emphasizing its importance (Optis et al., 2021; Hatfield et al., 2023).





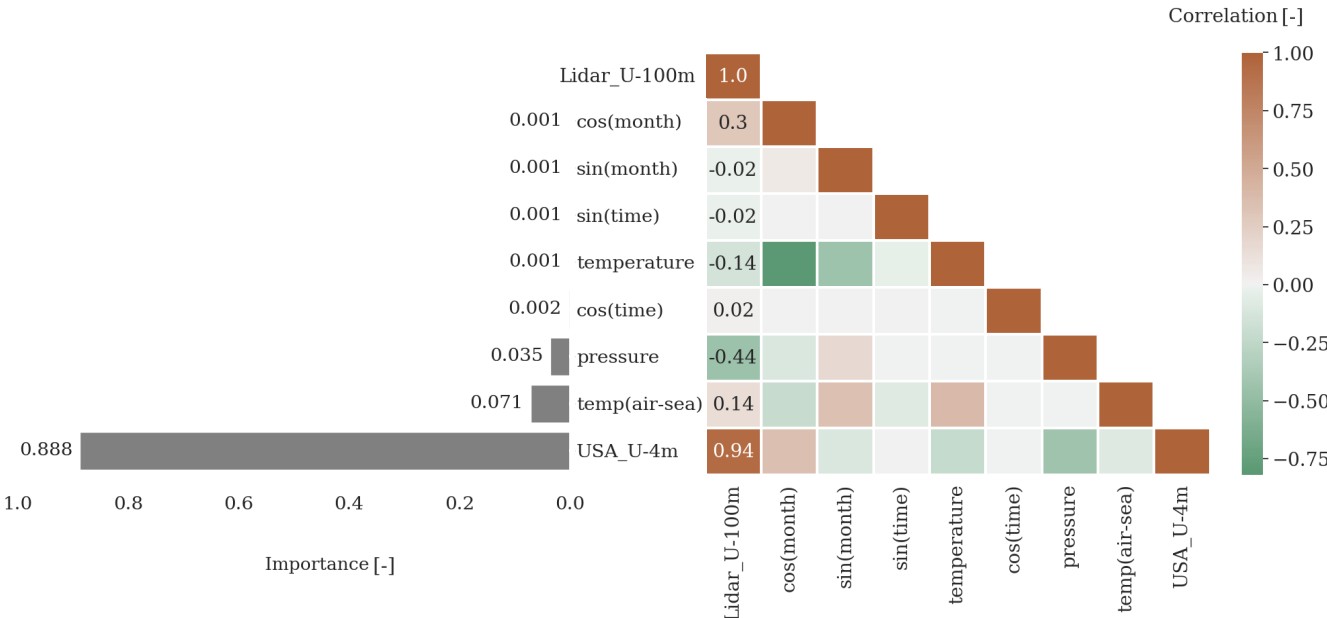

**Figure 3.** Importance of each input considered by the random forest regressor (left) and correlation of input variables with lidar-measured wind speed at 100 m (right). The data captured at HKW was used for these plots, and similar behavior was observed across different locations.

The alignment of feature importance with the correlation table, as illustrated in Figure 3 (right), provides a compelling indication of the random forest algorithm's effectiveness in identifying relevant variables and partitioning the data accordingly.

## 3.2 Wind speed reconstruction

The two random forest-based models, RF and QRF, were supplied with the aforementioned variables to predict wind speeds at elevations considerably exceeding the input heights (ranging from 30 to 200 m). Two distinct validation approaches were employed: same-site and round-robin.

In the same-site validation, the machine learning models were tested at the locations where they underwent training. The resulting wind speed profiles were benchmarked against corrected ERA5 profiles. This validation methodology ensures that the models are assessed in regions where they possess knowledge of the wind speed during the training time stamps.

Conversely, the round-robin validation involved a comparison against ERA5 outputs directly. In this case, the machine learning models were tested at locations spanning distances of 27 to 215 km from their training sites. The efficacy of the models in predicting wind speeds at locations significantly distant from the training sites is thus rigorously evaluated.





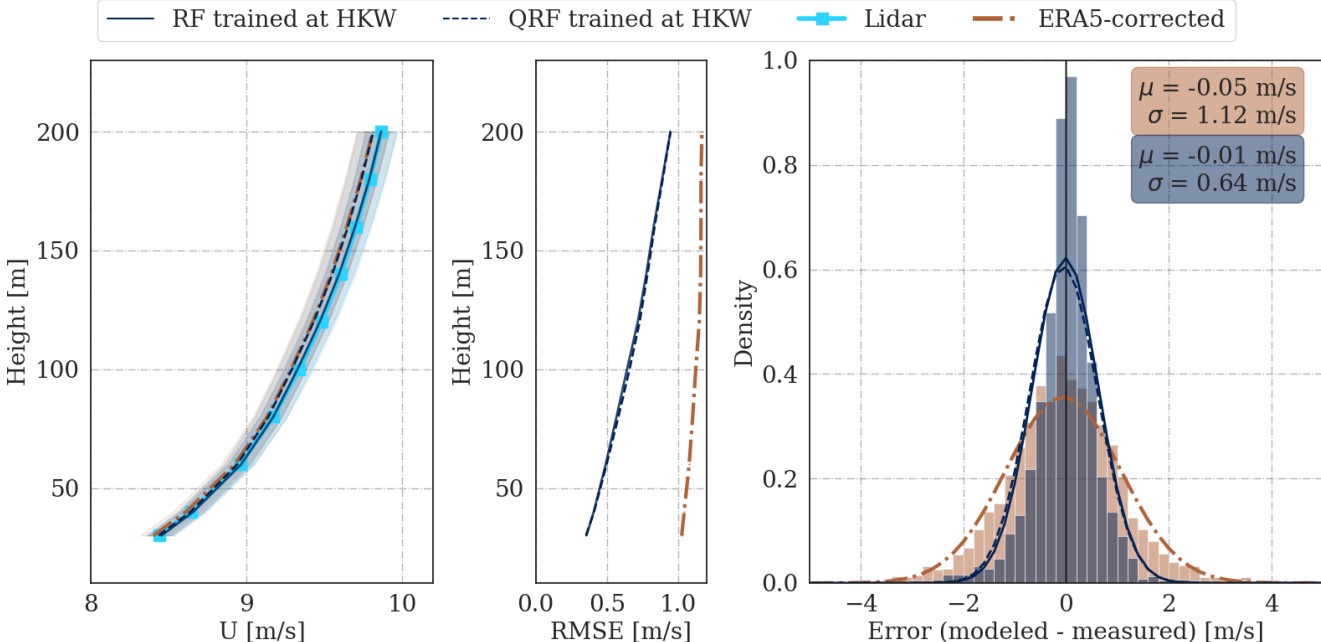

**Figure 4.** Same-site validation at the HKW location: average wind profiles (left), RMSE profile (middle), and the error probability density function (PDF) at 100 m (right) for the testing subset, comprising 2388 one-hour averages. Wind profiles are shaded with the standard error of the mean, and error histograms are binned into 0.2 m/s intervals.

### 3.2.1 Same-site validation

The wind profiles obtained through the same-site validation at HKW site are illustrated in Fig 4. In the left panel, the average profile predicted by the RF model exhibits a notable alignment with the lidar average profile. Conversely, the QRF model and corrected ERA5 profiles slightly underestimate the wind speed. To quantify this bias more concretely at 100 m, the right panel presents a Gaussian distribution fitted on the error density histogram. The mean of this distribution, representing the bias at 100 m, is -0.01 m/s for RF and -0.05 m/s for corrected ERA5. Notably, the random forest-based models demonstrate a narrower error distribution with a standard deviation of 0.64 m/s, indicating a 75% reduction compared to corrected ERA5. This narrower distribution contributes to a lower RMSE, as depicted in the middle panel.

The RMSE of machine learning (ML)-predicted wind speeds remains comparable for both RF and QRF, exhibiting an increase with height (50% from 100 to 200 m) where the conditions can become decoupled from surface-measured variables. This was also observed in prior studies (Bodini and Optis, 2020b; Hallgren et al., 2023). In contrast, corrected ERA5 shows a weaker dependency of RMSE on height, indicating consistent modeling of processes across all elevations. However, the RMSE of ERA5 wind speeds surpasses that of random forest-based models at all heights, averaging a 40% higher value. This





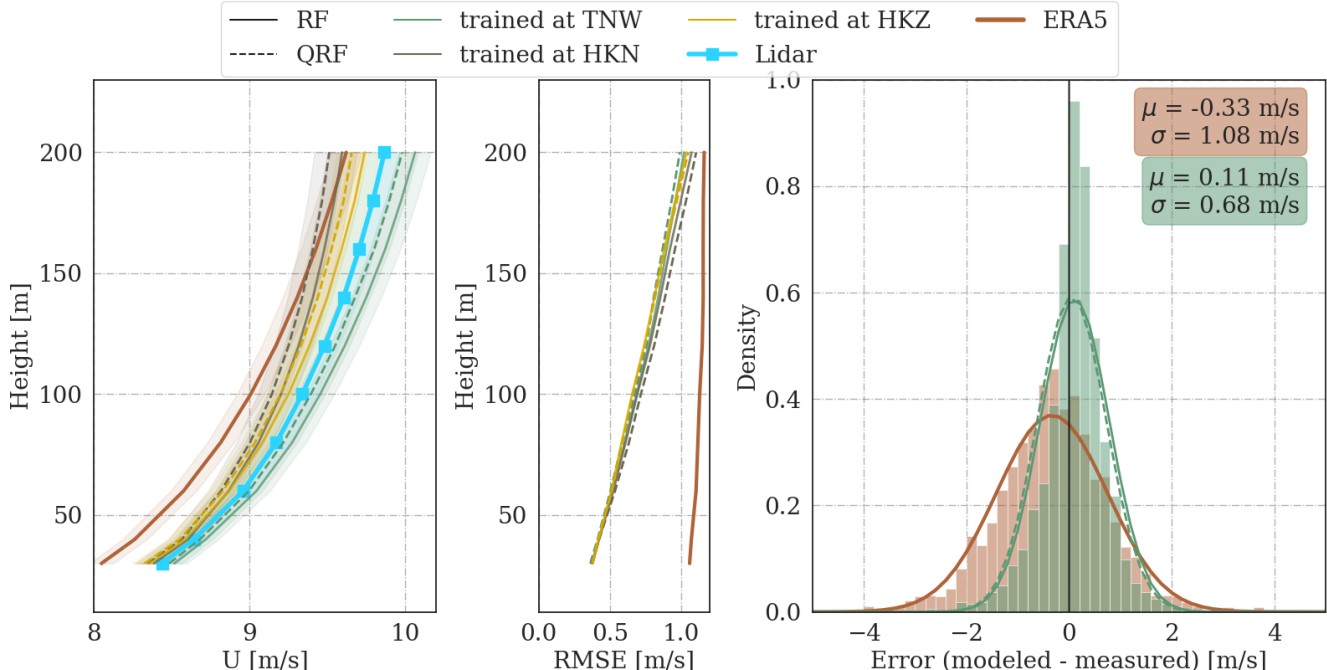

**Figure 5.** Round-robin validation at the HKW location: average wind profiles (left), RMSE profile (middle), and the error probability density function (PDF) at 100 m (right) for the testing subset, comprising 2388 one-hour averages. Wind profiles are shaded with the standard error of the mean, and error histograms are binned into 0.2 m/s intervals

underscores the potential of the random forest algorithm for filling the lidar gaps during a floating lidar campaign, given that
the met station collects data during the lidar gap.

### 3.2.2  Round-robin validation

In the round-robin validation conducted at HKW, the ML models trained at TNW, HKN, and HKZ were supplied with input variables at HKW, and their predicted wind speeds were compared with ERA5 outputs. The outcomes are depicted in Fig. 5. Notably, the QRF-predicted average wind speeds are consistently slower than those predicted by the RF model across all
training locations. Models trained at TNW exhibit a positive bias, growing with height, while those trained at HKN and HKZ display a negative bias. This discrepancy may be attributed to the higher average wind speeds at TNW, which are 0.32 and 0.74 m/s greater than those at HKN and HKZ, respectively (as per data presented in Table 2). The uncorrected ERA5 average profile exhibits a relatively constant negative bias for all heights, surpassing the bias of ML models in absolute value, except for ML models trained at HKN and at higher elevations (Fig. 5, left).
Examining the ML models trained at TNW and the ERA5 error distributions at 100 m (Fig. 5, right), it is evident that the spread of the ML error distribution is akin to the same-site approach (0.68 m/s), albeit with a slightly larger mean (12%). This marginal overestimation is noteworthy, considering the geographical distance between training and testing locations (202 km).





The mean of the ERA5 error distribution at 100 m is -0.33 m/s, which is three times larger in absolute value than that of the RF model trained 202 km away.

The RMSE of all round-robin ML models applied at the HKW site is very similar, on average 7% larger than the same-site approach. However, it is crucial to highlight that all round-robin ML models consistently outperform ERA5 in RMSE, indicating a significant improvement (37%). This underscores the robustness and potential of the proposed ML methodology in horizontally extrapolating wind profiles.

It is noteworthy that the MCP correction of ERA5 wind speeds ameliorated the bias to 0.07 m/s (by 61%) across all heights.
However, its impact on RMSE was limited (0.8%). Hence, we recommend considering the random forest models for site assessment calculations, if offshore near-surface measurements are available in the region, even at a distance of 200 km.

## 3.3    Error Analysis

To comprehensively evaluate the overall performance of the random forest model, three key performance metrics are presented in Fig. 6. The depicted metrics include Bias, Root Mean Square Error (RMSE), and the coefficient of determination ($R^2$).
These metrics collectively indicate a degradation in model performance as the application distance from the training locations increases. In the case of same-site validation (distance = 0), the Bias of the machine learning based models ranges between -0.09 and 0.04 m/s. However, it expands to -0.37 to 0.34 m/s when the models are trained 200 km away. While ERA5 systematically underestimates the wind speed in the North Sea (also noted by Dörenkämper et al. (2020)), even to the extent of -0.58 m/s. But with correction, the Bias lies between -0.07 and 0.07 m/s. It is advisable to maintain a minimum distance to training when
applying the random forest model to ensure a performance comparable to ERA5 in terms of Bias.

Additionally, RMSE is also observed to correlate with the distance to training, as it grows from 0.73 to 0.81 m/s, when the model is trained 170-215 km away. However, the random forest models trained at all distances consistently outperform ERA5 outputs, both before and after MCP correction.

Furthermore, the random forest models have an $R^2$ of 0.97 for same-site implementation, which drops slightly (0.2%) with
a training distance of 200 km. Nevertheless, even the ML models trained the furthest (up to 215 km away) outperform ERA5 by 4.4% in terms of $R^2$. It is pertinent to note that MCP correction does not alter the $R^2$ values due to its linear characteristics.





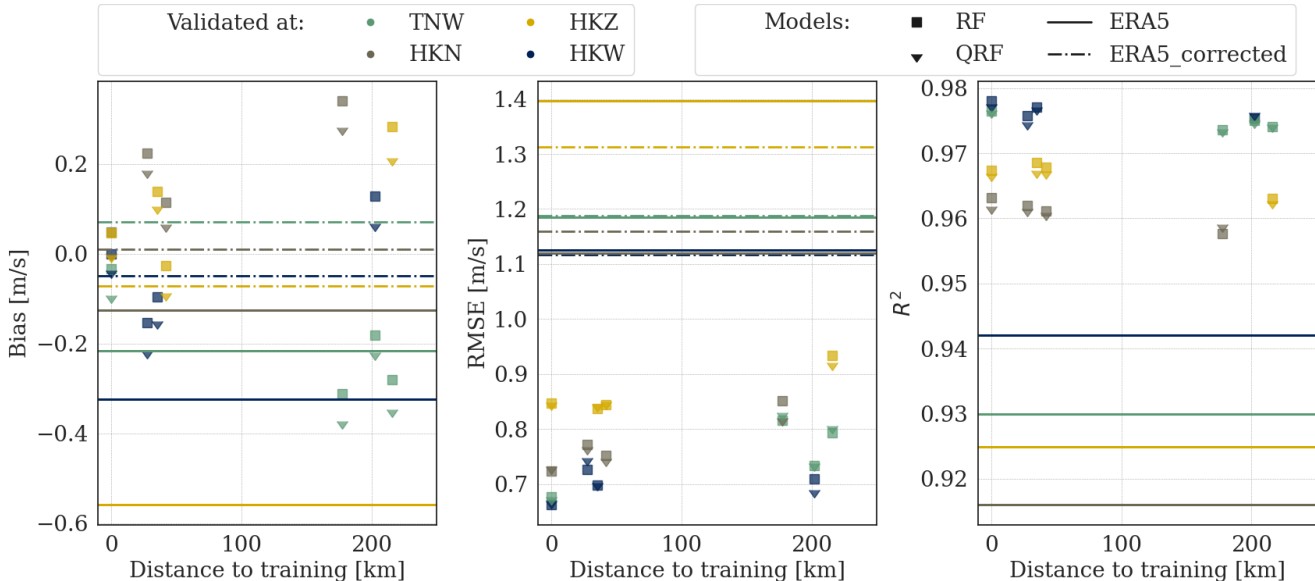

**Figure 6.** Dependence of errors on the distance between training and testing sites, with horizontal lines indicating ERA5 error metrics pre and post-correction at the 4 sites for comparison.

In Fig. 7, the dependency of Error on wind speeds and wind direction at HKW and 100 m is illustrated. The plot reveals that both the same-site and round-robin implementations of the RF model exhibit the highest precision and accuracy for wind speeds ranging from 4 to 20 m/s, a range frequently encountered in practice. The Mean Absolute Error (MAE) and Bias of
ERA5 appear to increase with wind speed. However, the Bias of ERA5 can be effectively corrected even for wind speeds up to 19 m/s.

Notably, both models demonstrate a higher MAE for winds coming from the southeast, corresponding to the coastal region. ERA5 has previously shown limitations in resolving coastal effects due to its spatial resolution (Rubio et al., 2022). The MAE in this wind sector also appears to be higher for the random forest models, suggesting challenges in modeling coastal effects.
This is the case for the TNW site with a different coastal direction as well (not shown). Nevertheless, the MAE in this section is 0.5 m/s lower than that of ERA5, indicating an improvement in performance with the random forest approach.

### 3.4 Wind speed variability

The ramp rate serves as a metric capturing the variability of wind speed over time. Hourly ramp rates were calculated to quantify the extent of wind speed changes within one hour. Datasets with higher temporal resolution, such as those measured
and modeled by random forest, were down-sampled to match the resolution of ERA5. Figure 8 illustrates a normal fit to the ramp rate distribution at the HKW site at 100 m for the testing samples.

Both the same-site and round-robin predictions of the random forest model exhibit good alignment with the measurements in terms of hourly variability. In contrast, ERA5 displays a narrower distribution compared to the lidar, indicating a deficiency





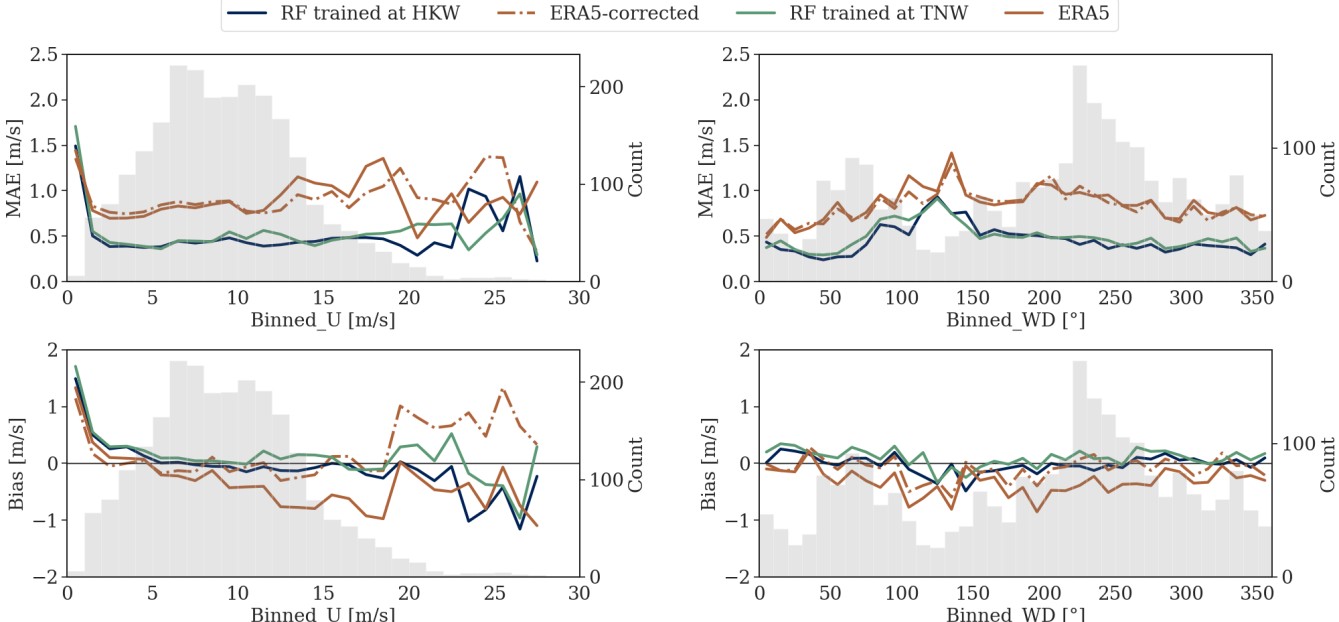

**Figure 7.** Error dependency on wind speed and direction at 100 m, measured by lidar at the HKW location. Error metrics are binned into 1 m/s and 10° intervals for the left and right panels, respectively. The background shading illustrates the distributions of measured wind speed and direction in gray.

in properly modeling hourly variability. Notably, after MCP correction, the distribution becomes marginally wider, bringing it closer to the measurements.

The ramp rate results are quantified in Table 4, where $\mu$ is the mean of the absolute ramp rates and $\sigma$ the standard deviation of the Gaussian distribution, shown also in Fig. 8. ERA5 hourly wind speed variability shows an underestimation of 27% and 28% on mean and standard deviation, which reduces to 22%, when MCP correction is incorporated, indicating an improvement in ERA5's ability to capture hourly wind speed changes.

Contrastingly, the random forest algorithm appears to accurately model wind speed variability with only a 4% to 5% overestimation for both the same-site and round-robin approach. The low spatial and temporal resolution of ERA5 may contribute to the observed larger deviation in mean and standard deviation of ramp rates. This emphasizes the impact of spatial grid characteristics on the variability representation within ERA5. The lower deviation in RF might be influenced by the fact that RF models are trained on point measurements, leading to a more localized understanding of wind variability.



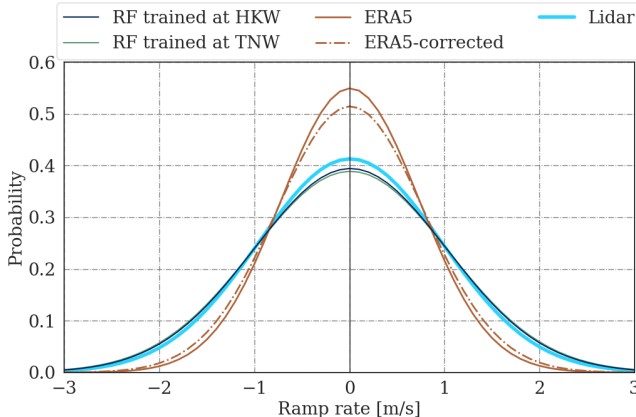

**Figure 8.** Gaussian fit to the hourly ramp rate distribution at the HKW site, here for 100 m.

**Table 4.** Ramp rate statistics at 100 m: $\mu$ corresponds to the average of the absolute change of the wind speed in a one-hour period. $\sigma$ is the standard deviation of the normal fit to the ramp rate distribution.

| | | | Deviation (%) | | | |
|---|---|---|---|---|---|---|
| | | | Same-site | | Round-robin | |
| Site | Ramp rate stats [m/s] | Obervation | RF | ERA5_corr | RF | ERA5 |
| HKW | $\mu$ | 0.69 | 3.3 | -20.0 | 2.1 | -25.1 |
| | $\sigma$ | 0.97 | 4.7 | -19.8 | 3.3 | -24.8 |
| HKN | $\mu$ | 0.74 | 6.3 | -27.0 | 9.3 | -30.1 |
| | $\sigma$ | 0.98 | 7.8 | -26.6 | 10.7 | -30.3 |
| HKZ | $\mu$ | 0.77 | 4.5 | -23.2 | 4.8 | -29.7 |
| | $\sigma$ | 1.08 | 4.4 | -24.2 | 4.6 | -31.3 |
| TNW | $\mu$ | 0.76 | 4.5 | -16.8 | 1.4 | -22.2 |
| | $\sigma$ | 1.06 | 6.1 | -19.5 | 2.3 | -24.7 |

## 3.5 Spectral analysis


Although it deviates from reality, scientists often approximate the wind speed signal as the superposition of waves with different frequencies to gain a physical understanding of eddy sizes. We conducted a comparison between the measured power spectral density and the modeled ones at the TNW site at 100 m, as depicted in Figure 9. The plot represents 2 years of data at TNW (not a testing subset). Consequently, only predictions of the random forest model developed at HKW are shown and cross-compared

with the non-corrected ERA5 predictions.





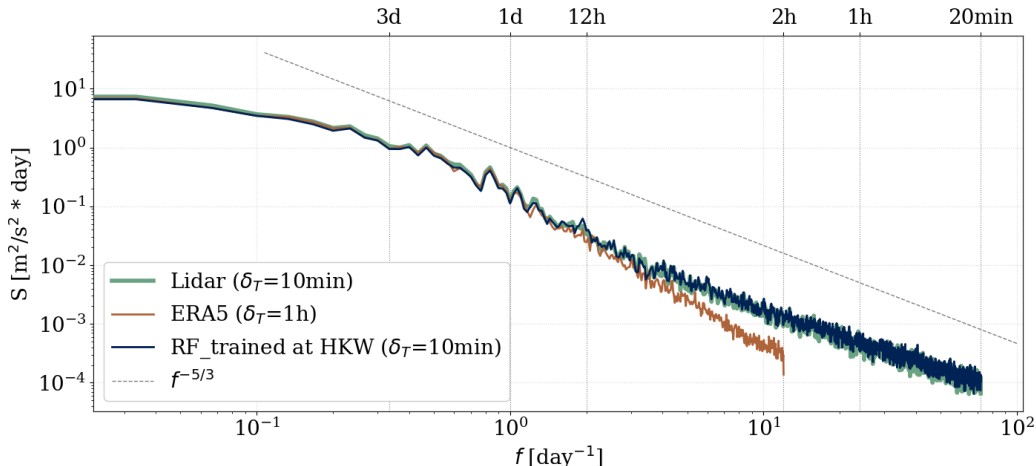

**Figure 9.** Power spectral density at 100 m at TNW, calculated as described in section 2.5.3. $\delta_T$ represents the temporal resolution of the dataset and the $-\frac{5}{3}$ slope from the Kolmogorov law ($S \propto f^{-\frac{5}{3}}$) in the inertial subrange of turbulence is also shown in gray (Kolmogorov, 1941).

As previously mentioned in section 3.4, ERA5 faces challenges in capturing wind speed variability due to its resolution. In fact, we do not expect ERA5 to resolve the inertial subrange of the spectrum thoroughly, given its resolution. Figure 9 confirms this, illustrating a degradation of energy for higher frequencies but good alignment for low frequencies, also shown by Meyer and Gottschall (2022). On the other hand, the random forest model trained 202 km away from TNW was able to capture eddies

as small as the lidar could.

### 3.6   Atmospheric stability

In the final phase of our investigation, we focused on evaluating the performance of machine learning models under various stability regimes. The upper panels of Figure 10 showcase the averaged error metrics across all heights for both random forest models, encompassing all locations. Notably, machine learning models demonstrate increased accuracy in predicting wind

speeds during unstable conditions, consistent with findings from Optis et al. (2021) and Bodini and Optis (2020b). Comparing stable and unstable conditions, we observe that the RMSE and Bias are 12% and 29% larger, respectively, for stable conditions. This discrepancy is attributed to the decoupling effect at higher heights from the surface, influenced by lower turbulent fluxes in the vertical direction, during stable stratification.

Moving to the lower panels of Figure 10, we delve into a comparative analysis of the two validation approaches under

distinct stability regimes. As detailed in section 3.3, the round-robin approach consistently yields less accurate results than the same-site validation, irrespective of the stability regime. Interestingly, the decline in accuracy when applied away from the training location cannot be solely attributed to atmospheric stratification, as the proportionate decrease in accuracy remains



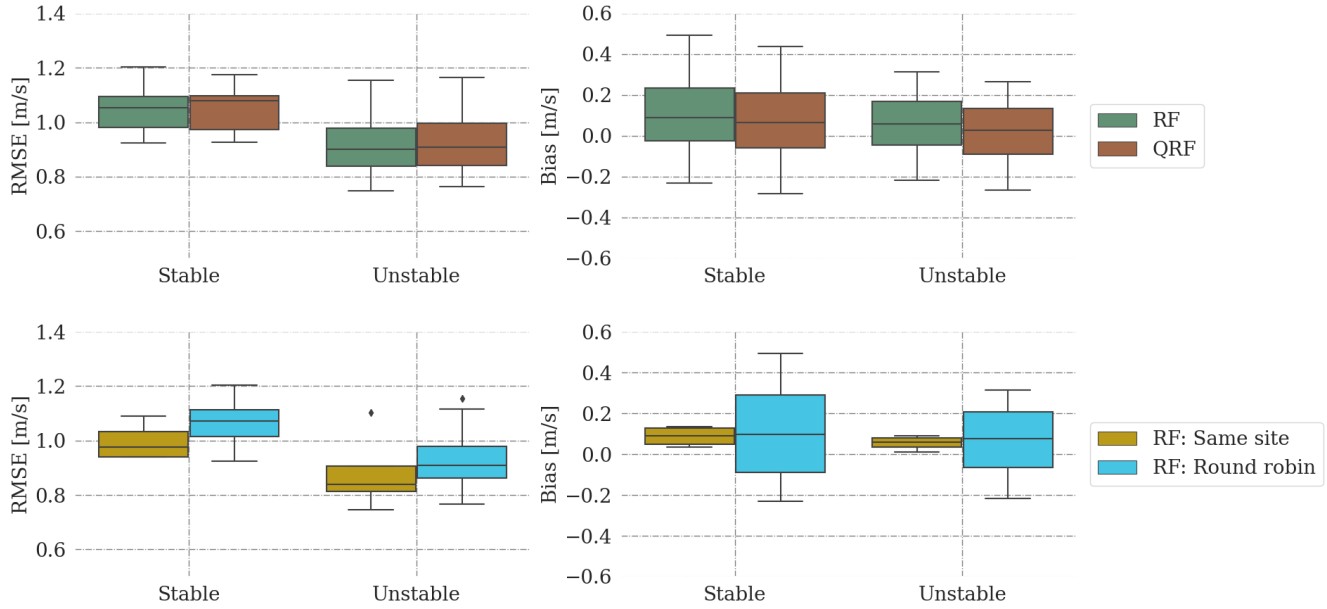

**Figure 10.** Box plot depicting the average error metrics for stable and unstable conditions across all heights, with variations attributed to different locations.

consistent for both stable and unstable conditions. The transition to the round-robin approach introduces an increase in bias in absolute value, with a slightly more pronounced effect observed for stable conditions.

# 4 Discussion

In this study, we investigated the performance of the random forest algorithm on vertical extension of buoy wind profiles. We laid our focus on how accurate the random forest can predict the wind speed variability and capture different structures, where ERA5, commonly used for site assessment, shows inefficacy (Dörenkämper et al., 2020; Meyer and Gottschall, 2022). We proposed a methodology to assimilate near-surface measurements into a model based on two machine learning algorithms: the random forest regressor (RF) and quantile regression forest (QRF). While RF is a well-researched method, QRF is introduced to the wind energy literature for the first time. QRF predictions generally trend slightly lower than those of RF, exhibiting a similar performance when validated against lidar measurements. The advantage of QRF lies in reporting the desired uncertainty as a byproduct, albeit at the cost of a longer modeling time. For benchmarking, we chose the ERA5 dataset, and corrected it using the "Measure, Correlate, Predict" (MCP) method, a common method for long-term extrapolations in site assessment calculations.





The top environmental variables that the wind speed aloft correlates with are: near surface wind speed, air-sea temperature difference and pressure. This is well captured by the random forest, as these have the highest impact on minimizing the least squared error (see Fig. 3). The cosine of month and the air temperature also correlate well with the wind speed at 100 m, but are not considered important by random forest. This can be due to the fact that they are not independent of the ones considered

important. For instance, the dependency of temperature on pressure can be described by the ideal gas law. Pressure is considered more helpful, most probably because it changes more drastically than temperature. The air-sea temperature difference, used as proxy for atmospheric stratification, contributes to 34% RMSE improvement. This aligns with previous findings by Hatfield et al. (2023) and Optis et al. (2021), both of whom demonstrated that excluding this variable can result in up to a 20% increase in RMSE. The inclusion of near surface wind direction as an input variable has demonstrated efficacy in enhancing model

performance during testing at the same-site as training (not shown). However, a nuanced consideration is imperative, as its applicability becomes potentially misleading, when extrapolated to locations characterized by a coastal direction differing from that of the training site. Since the model is trained without explicit information about changes in coastal direction, it may not effectively account for such variations in different geographical contexts. In order to safeguard the universality and robustness of our model within the designated domain, a deliberate decision has been made to exclude wind direction as an

input variable.

Our analysis demonstrated that, in cases where wind profiles exist at a specific location (for training), machine learning models fed with the near-surface measurements outperform corrected ERA5 profiles, with a 39% improvement in RMSE and a 35% improvement in Bias. This is a fair comparison, where both models have knowledge of wind speeds up to 200 m at the site. This improvement aligns with a study by Schwegmann et al. (2023), where the top 5 machine learning models, including

random forest, demonstrated superior performance, outpacing MCP-corrected ERA5 wind speeds by up to 28% in RMSE. These findings underscore the potential value of a random forest model in filling measurement gaps during offshore campaigns in the pre-deployment phase. Such gaps, arising from system failures (e.g., of sensors, the electronic cabinet, or power supply) or low backscatter, are more common in offshore locations with challenging weather conditions.

The round-robin validation, initially proposed by Bodini and Optis (2020b) to avoid an overestimation of machine learning

performance, revealed that horizontal extrapolation of ML models has an adverse effect, particularly on Bias. Bias grows up to 0.28 m/s (300%) in absolute value with a 200 km distance from the trained location, shown in Fig. 6. Hence, we recommend maintaining a minimum distance to the training location to ensure Biases comparable with corrected ERA5. However, all round-robin validations yield lower RMSE and higher $R^2$ than ERA5, both pre and post-correction. A dataset with a larger Bias but higher accuracy may be preferred, as the systematic Bias can be removed using post-processing methods. Notably, the

performance improvement of random forest compared to ERA5 declines with height, as also found by Hallgren et al. (2023).

In conditions where the wind aloft is decoupled from the surface, as observed during stable stratification, the prediction of wind speed poses increased challenges for the random forest model, primarily due to its reliance on information from the near-surface level (see Fig.10). A similar observation was made by Optis et al. (2021).

One notable finding of this study was that ERA5 encounters challenges in modeling winds from coastal areas. Our analysis

reveals that the random forest model also exhibits the highest MAE in the coastal wind sector but is still 0.5 m/s more accurate





than ERA5 (see Fig. 7). As a next step, a more in-depth investigation could explore the feasibility of incorporating coastal characteristics as additional input variables into the model. This inquiry may lead to further refinements in the random forest model's performance, particularly in regions influenced by proximity to the coast. We also observed the largest negative Bias of ERA5 at higher wind speeds, effectively removed using MCP. Nevertheless, the random forest model trained 200 km away

showed a lower Bias and MAE than both ERA5 and corrected ERA5 for almost all wind speed ranges at the HKW site.

Our main objective was to assess the potential of employing random forest to address the underestimation of wind speed variability in ERA5, linked to its coarse spatial grid (Meyer and Gottschall, 2022). Our findings confirm that ERA5 consistently underestimates wind speed variability, exhibiting a deviation of 22-30% from observed absolute wind speed hourly changes (refer to Table 4). After correction, this deviation reduces to 16-27% across all sites. In contrast, the machine learning method

proves more accurate, demonstrating a deviation of 2-9%. The limited resolution of ERA5 also impedes its ability to capture the inertial subrange of the power spectrum. Remarkably, the random forest model, even when trained 200 km away, successfully models eddies as small as those observed in the measurements. This capability stems from its training on 10-minute point measurements, rendering it more localized and adept at capturing fine-scale wind patterns.

Our analysis deliberately excluded wake effects by filtering out the influence of neighboring wind farms to maintain the

model's simplicity, allowing a focused analysis to attribute errors accurately. However, as wind parks become more concentrated in the North Sea area, and long-lasting cluster wakes become prevalent, the prospect of incorporating wake effects into the modeling process and validation presents itself as an intriguing avenue for future exploration. This potential advantage gains significance, particularly as current state-of-the-art reanalysis data lack comprehensive coverage of the wind farms. The inclusion of wake effects in future research could significantly enhance our understanding of wind behavior in regions with

dense wind park concentrations. Realizing this potential would greatly benefit from a multinational network of floating lidars in the North Sea, provided that the collected data is made publicly available. To achieve this, international collaboration is essential.

The results of this study demonstrate that utilizing a random forest model trained at the North Sea, coupled with a network of buoys equipped with redundant met stations, has the potential to significantly enhance the existing estimations of wind

profiles. This improvement can greatly benefit site assessments, leading to substantial cost savings, as uncertainties in wind speed estimations directly propagate into financial uncertainties. It is essential to underscore that, even with the implementation of the machine learning model, the need to measure the wind profile will not be entirely eliminated. The machine learning model, being fundamentally data-driven, must undergo thorough validation against measurements to ensure its reliability.

## 5    Conclusion

The accurate determination of wind speed at the height of the rotor swept area is crucial for effective site assessment and precise yield calculations in the wind energy industry. While the industry commonly relies on integrating ERA5 data with short-term measurements through the MCP method, our study brings attention to ERA5's limitations in capturing site-specific wind speed variability due to its inherent low resolution.



To address this challenge, we developed random forest models capable of extending near-surface wind speed up to 200 m. Our study focused on the Dutch part of the North Sea, where we meticulously trained, cross-compared, and verified these models using floating lidars from four sites.

Our findings underscore the superiority of the random forest model when provided with wind profiles at the site. In such cases, it outperforms MCP-corrected ERA5 wind profiles in terms of accuracy, bias, and coefficient of correlation. However, when wind measurements at the rotor height are unavailable, a model trained on wind profiles within a 200 km region can effectively handle the vertical extension. Our analysis reveals that the horizontal extension of the model primarily impacts bias, with a significant increase when the training location is 200 km away, reaching up to 0.37 m/s in absolute value (300%), accompanied by a relatively minor drop in accuracy.

Notably, a random forest model trained on local wind profiles demonstrates superior accuracy in capturing wind speed variations and local effects. Compared to the corrected ERA5, which exhibits a 22% deviation in absolute hourly wind speed variability at a site, a random forest model trained at a distance shows an average deviation of up to 5%. Additionally, the machine learning model adeptly captures the inertial subrange of the power spectrum, where ERA5 experiences degradation.

Our results highlight the potential of machine learning methods and offer a feasibility analysis for their use in resource assessment. Future research could explore extending the random forest methodology to extrapolate the wind profile above 200 m, potentially benefiting the next generation of offshore wind turbines. Additionally, as the North Sea progresses toward becoming one of the world's most concentrated offshore wind energy regions, there is a compelling need to investigate cluster wakes. This future area of study suggests the incorporation of wakes into these models, with a multinational network of floating lidars in the North Sea playing a crucial role, provided that the data is made publicly available. (Cañadillas et al., 2022)

*Data availability.* Floating lidar campaigns were performed by Fugro, and the data was made publicly available by Netherlands Enterprise Agancy (Rijksdienst voor Ondernemend Nederland, RVO) at https://offshorewind.rvo.nl/ (last access: 23 August 2023). The ERA5 data on pressure levels are available via the Copernicus Climate Change Service (C3S) from https://cds.climate.copernicus.eu/ (last access: 12 October 2023).

*Author contributions.* F.R contributed to the preprocessing of the floating lidar datasets, implemented the random forest methodology, performed the cross-comparisons and wrote the manuscript. J.G. contributed to the funding acquisition, conceptualization of the study, and participated in numerous discussions, and provided a comprehensive review of the manuscript.

*Competing interests.* At least one of the (co-)authors is a member of the editorial board of Wind Energy Science.



*Acknowledgements.* We express special gratitude to Pedro Santos for his significant contribution and support in the early stages of this work. Additionally, we appreciate the valuable insights provided by Ashim Giyanani, Lin-Ya Hung, and Hugo Rubio.

# Appendix A

## A1 Bulk Richardson number derivation

Equation 8 adopted from Stull (1988) was used to derive the bulk Richardson number using parameters detailed in section 2.6. The set of Equations (A1 - A7) shows how the bulk Richardson number is obtained form the measured variables.

We assumed the humidity stays constant within the first 4 m above sea level. To extrapolate pressure to sea surface (SS), the barometric formula is used, which allows for temperature changes in the extrapolated height (Lente and Ősz, 2020):

$$P_{SS} = P_{4m}(\frac{T_{SS}}{T_{4m}})^{\frac{Mg}{\gamma R}}, \tag{A1}$$

where $M = 0.0289 \frac{kg}{mol}$ is the molar mass of air, $R = 8.314 \frac{J}{mol \cdot K}$ is the universal gas constant and $g = 9.81 \frac{m}{s^2}$ is the gravitational acceleration. $\gamma$ is the temperature gradient between sea surface and 4 m:

$$\gamma = -\frac{T_{4m} - T_{SS}}{4}. \tag{A2}$$

One of the main parameters in calculating the Ri is the virtual potential temperature. The virtual temperature accounts for the water vapor in the air parcel and the changes it brings due to its lower density. Hence it allows to use the equation of state,

which is valid for dry air. It is described by:

$$\theta_v = \theta(1 + 0.61r). \tag{A3}$$

The potential temperature accounts for the variations due to the pressure difference. It is defined as the temperature that air would have if brought to a reference pressure ($P_0 = 100 \ kPa$) through an isentropic process (Stull, 1988):

$$\theta = T(\frac{P_0}{P})^{\kappa}, \tag{A4}$$

where $\kappa$ is the Poisson constant and can be approximated by $\kappa \approx 0.2854(1 - r)$ for moist air. Here $r$ is the water vapor mixing ratio and can be obtained by:

$$r = \frac{R_d}{R_v}(\frac{e}{P - e}), \tag{A5}$$

where $R_d = 287.05 \frac{J}{kg.K}$ and $R_v = 461.52 \frac{J}{kg.K}$ are the specific gas constant of dry air and water vapor, respectively. $e$ is the vapor partial pressure, and can be derived from the relative humidity:

$$e = \frac{RH}{100}e_s, \tag{A6}$$

where $e_s$ is the vapor pressure at saturation, and is approximated by the Clausius-Clapeyron equation as a function of temperature (Iribarne and Godson, 1973; Bolton, 1980):

$$e_s = 611.2exp(\frac{17.67(T - 273.15)}{T - 29.64}) \tag{A7}$$



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
