# Peer review of "Characterization of Local Wind Profiles: A Random Forest Approach for Enhanced Wind Profile Extrapolation"

_Wind Energy Science, 2023_

## Author Response (AR1)

Dear Reviewers,

Thank you for your insightful comments. We have undertaken a detailed revision based on your suggestions. In the following pages, you find our responses to your comments in blue and the tracked changes made in the subsequent section. We believe your contributions have improved the quality of the manuscript considerably, and we appreciate your effort.

Best regards,
Farkhondeh Rouholahnejad
on behalf of the authors

**Referee 1**

1. In the abstract (Line 9), there is no limit on the collection time and sample size of comparative data, only indicating that the comparative results are meaningless. In addition, the analysis of wind speed variability, spectral analysis, and atmospheric stability in the results section was not mentioned in the abstract.

   We adjusted line 5 to: "Based on public 2-year floating lidar data collected at four locations, the 15% testing subset shows that the random forest model trained on the remaining 85% of site-specific wind profiles outperforms the MCP-corrected ERA5 wind profiles in accuracy, bias, and correlation".

   The wind speed variability improvement using a random forest model was mentioned in "Our regionally trained random forest model exhibits superior accuracy in capturing wind speed variations and local effects, with an average deviation below 5% compared to corrected ERA5 with a 20% deviation from measurements."

   The spectral analysis was mentioned in the old abstract and we adjusted the sentence to "The 10-min random forest predicted wind speeds capture the meso-scale section of the power spectrum where ERA5 shows degradation." to be clearer.

   We added the information regarding the stability in the revised abstract (For stable conditions the RMSE and Bias are 12% and 29% larger, respectively compared to unstable, which can be attributed to the decoupling effect at higher heights from the surface during stable stratification).

2. In section 2.1, as the author stated, "A reliable dataset is key to train and validate a data driven model". Here should be an explanation of the accuracy of the LiDAR wind profile product.

   According to the pre-deployment verification test at Frøya, the floating lidar systems were compared against land-based lidar systems over a period of 36.7 days (DNVGL, 2019). The Seawatch system has passed the acceptance criteria provided by the carbon trust roadmap for commercial acceptance of Floating LiDAR with a $R^2$ higher than 0.97 and a slope of a linear regression in a range of 0.099 to 0.0127 for the wind speed. We added verification report to the reference list and mentioned it in section 2.1 ("These systems undertook

verification tests, and passed the acceptance criteria provided by the roadmap for commercial acceptance of Floating LiDAR").

3. In section 2.3, only 15% of the sample size was used for training and validation. Lack of explanation on the total number of training samples. Additionally, in subsequent inversion and comparison, did the sample point use hourly average wind speed data or minute level data?

   We adjusted table 2 to add the number of samples for each location.

   In section 2.3 we have "The random forest models' outputs and the measured data are down-sampled to match the temporal resolution of ERA5 and stamped in the middle of the period". We adjusted it to "down-sampled to one hour to match the temporal resolution of ERA5". However, the spectral analysis is based on the original temporal resolution (10 min for random forest and lidar, 1 h for ERA5), as obvious from the highest frequency in Figure 9.

4. In section 2.6, atmospheric state partitioning is based on the Richardson number for subsequent analysis. Here, it should be clarified how many samples are stable conditions and how many samples are unstable conditions for each site.

   We added Figure 10 in the revised manuscript, which shows the 1/L distribution for each site and provides an overview about the number of stable and unstable samples. We adjusted section 3.6 accordingly.

5. There are no sub image numbers such as (a), (b), and (c) in all the images, making it difficult to understand what results each sub image corresponds to. Taking Figure 10 as an example, do the four subgraphs correspond to four sites?

   We adjusted the figures to separate the upper and lower panels for clarity.

6. Lines 39 and 57: The reference format is incorrect. "Bodini and Optis (2020a)" is supposed to appear first. In addition, there are some pioneering papers that are highly relevant to this manuscript, which should not be ignored. I just list several references as follows:
   a. https://doi.org/10.1016/j.rser.2022.112897

b. https://doi.org/10.1073/pnas.2119369119;
c. https://doi.org/10.5194/acp-23-3181-2023;
d. https://doi.org/10.5194/egusphere-2023-2727.

Bodini and Optis (2020b) lays more focus on the round robin validation, and hence cited in the paragraph talking about this validation approach. Later on, when the potential of random forest in capturing wind events is discussed, we mentioned Bodini and Optis (2020a).  Although this paper uses the round robin approach, but it focuses on the accuracy of random forest in different wind conditions, and hence fits the context.

Thank you for your comment. We adjusted the literature study (section 1 ) to include the proposed papers.

**Referee 2**

1. How does the proposed method compare to ERA5-corrected method in terms of efficiency? ML methods normally take a considerable amount of time to train, but a typical MCP process using ERA5 data can be very fast.

   We added the following sentences to the discussion: "The MCP method is known to have low computational effort, which can offer advantages over more computationally intensive ML methods. The computation time for the Random Forest model is highly dependent on hyperparameters, particularly the number of trees. In our study, the training time per location ranged from 21 to 77 seconds, demonstrating the model's computational efficiency."

2. The hyperparameters listed in Table 3 for HKW, HKN and HKZ are different. To my limited understanding towards ML methods, hyperparameters act as choice of models/methods in numerical simulations. For example, both RANS equations and LES can be solved to numerically simulate a fluid flow, the choice between them depends on flow physics for a specific problem. Towards this end, the difference in hyperparameters may indicate different wind conditions among these sites. Would the authors make some comments on this?

   We added the following paragraph to the discussion to provide the physical interpretation of the hyperparameters: "The hyperparameters of the random forest algorithm determine the structure and quantity of trees in the model. These parameters set the convergence criteria for terminating data splits and

subsequently using the mean value of the population within a leaf node as the predictive output. When the input features effectively capture the underlying system dynamics, a shallow tree structure can yield accurate predictions. Upon visualizing a specific decision tree, we observed a clear stratification where timestamps associated with higher wind speeds predominantly occupied one side of the tree, while lower wind speeds were clustered on the other side. This outcome is likely influenced by the frequent utilization of the 4 m wind speed feature within the algorithm. In regions characterized by unstable stratification, it is plausible that the decision trees are inherently shorter, given that surface wind speed may serve as a robust indicator of the wind profile. Despite the absence of explicit physical modeling within the machine learning framework, the algorithm demonstrated an ability to organize the data in a manner consistent with established physical principles."

3. In Figure 4, the left plot shows the wind profile predicted using ERA5-corrented data, but in Figure 5 ERA5 data is used for comparison. What is the difference between "ERA5-corrected" data and "ERA5" data? Also, do both cover the same period shown in Table 3?

   For Figure 4, we plotted the performance of random forest and ERA5 on the validation subset at HKW. The random forest was trained at HKW, so it has prior knowledge about the location. By correcting the ERA5 profile using the same subset we used to train the random forest, we make sure that the comparison is fair.

   For Figure 5, the random forests are trained on other sites, and were not informed about the wind profiles at HKW. Hence it is fair to compared them directly ERA5, no correction applied.

   The same validation period is used for both Figure 4 and 5 (The lidar is the same curve). Only the models are different.

   We added "In this study, the corrected ERA5 wind profiles are referred to as ERA5-corrected." To the end of section 2.4, to make it clearer to the reader.

4. In Figure 9 "ERA5" data is used, but as is stated in the manuscript, it is expected that ERA5 data is not able to resolve the inertial subrange of the power spectrum. It would be much more interesting to show how ERA5-corrected data perform in this scenario.

   The MCP method primarily removes the bias from the dataset. As also shown in the middle panel of Figure 4 and 5, the RMSE of ERA5 does not change much

after the correction. We plotted the PSD for ERA5 both before and after correction at TNW location for the whole measurement period. We used 85% of the profiles to derive the MCP parameters and applied them on the entire period. As shown in the plot, only for lower frequencies there is a deviation between the corrected and uncorrected ERA5, which indicates that only the bias is removed when the MCP correction is used.  We added the following to the discussion to address this matter: "Our analysis demonstrated that the MCP correction applied to the ERA5 predictions does not mitigate the degradation at higher frequencies (data not shown) but primarily alters the energy at lower frequencies, thereby correcting the bias."

[Figure]

5.  In Figure 4 and 5,  shaded area (denotes standard error of the mean) and line (denotes the mean) with same color are used to represent results from one specific model, but their colors are not easy to distinguish from one and another.

    We adjusted the color of one of the models in the new version for clarity.

6.  Also, lidar measurements at different heights are connected with lines, which indicates a certain relation between these measurements. However, rigorously speaking this relation is not known. This principle also applies to plotted lines in the left panel in Figure 4 and 5 (which shows random forest model results). Do wind speed for each height is predicted (e.g., with 1m interval) or only some specific heights are predicted (e.g., heights listed in Table 1)?

    Thank you for the comment. The random forest predicts the wind speed at the heights it was trained on, which is the lidar heights. We corrected the plots in

Figure 4 and 5 to have the markers on the measurement heights for the random forest profiles. The ERA5 was interpolated, hence we leave it without markers.

7. In Figure 7 only RF trained results are shown, what about QRF ones?

The QRF predictions are very similar to RF ones for all sites. Here an example of the binned error metrics at TNW location for the RF and QRF trained at HKW:

[Figure]

We added the sentence "The results were similar for the QRF model." to section 3.3.

**References**

DNVGL. (2019). *Assessment of the Fugro Seawatch Wind LiDAR Buoy WS 191 Pre-Deployment Validation at Frøya, Norway*. Retrieved 12 19, 2023, from https://offshorewind.rvo.nl/file/download/26ae4742-2148-4748-ab4a-7a56a961b982/1575890079tnw_20191209_mc_validation%20ws191-f.pdf

---

## Author Response (AR2)

Dear Julie Lundquist,

We believe that the required corrections enhance the quality of the work, and we appreciate your contribution. Here we address the three issues you have raised:

1) *On a scientific point, the definition of atmospheric stability bins requires some justification; please provide some references in at the end of Section 2 to justify your choice of |L| ~ 1000 m as the dividing line between stable/unstable and neutral (perhaps Muñoz-Esparza et al 2012 could be helpful here).*

Thank you for your comment regarding the atmospheric stability bins.

In literature, classifications based on the Obukhov length (L) vary slightly. For example, Hansen, 2012 defined very stable and stable conditions with ranges of 10<L<50 and 50<L< 200, respectively, while ranges of -100<L<-50, -200<L<-100 were used for very unstable and unstable conditions. However, many studies adopt |L| <200 for very stable/unstable and 200<|L|<1000 for stable/unstable conditions (e.g., Motta, 2005; Watson, 2014).

Since the primary purpose of our stability classification is model error characterization, we did not differentiate between very stable (unstable) and stable (unstable) conditions. Therefore, we chose |L|=1000 as a practical threshold for defining stable/unstable and neutral categories, as also used by Schneemann Schneemann, 2021. We have added the following to Section 2.6:

**"The choice was based on literature conventions, where ranges such as 0<L<200 (-200 < L < 0) and 200<L<1000 (-1000 < L < -200) are common for very stable (very unstable) and stable (unstable) conditions (Argyle and Watson, 2014; Motta et al., 2005). In this study, we do not distinguish between stable (unstable) and very stable (very unstable) conditions, as we use this classification primarily for error characterization."**

2) *One reviewer pointed out that figure panels are not separately labelled, which makes it difficult for readers to understand exactly which panel is being discussed. I note that the author instructions state "Labels of panels must be included with brackets around letters being lower case (e.g. (a), (b), etc.).", so please revise to include panel labels, and you may also use the panel labels throughout your discussion to improve clarity. Specifically, Fig. 7, 11 need attention in this regard (Fig 10 is fine).*

We updated the Fig. 7 and 11 in the new version and adjusted the text accordingly.

3) *Some of the figures are not friendly to color-blind viewers. As recommended at https://www.wind-energy-science.net/submission.html#figurestables, please run your figures through https://www.color-blindness.com/coblis-color-blindness-simulator/ and modify*

*accordingly (I note that Fig 3 color loses meaning for Red-Blind/Protanopia, so you should check others as well.*

We ran all the figures through https://www.color-blindness.com/coblis-color-blindness-simulator and updated the plots, making changes to the colors, when necessary.

**References**

Hansen, K. B. (2012). The impact of turbulence intensity and atmospheric stability on power deficits due to wind turbine wakes at Horns Rev wind farm. *Wind Energ.*, 183-196. doi:https://doi.org/10.1002/we.512

Motta, M. a. (2005). The influence of non-logarithmic wind speed profiles on potential power output at Danish offshore sites. *Wind Energy*, 219-236. doi:https://doi.org/10.1002/we.146

Schneemann, J. a. (2021). Offshore wind farm global blockage measured with scanning lidar. *Wind Energy Science*, 521-538. doi:10.5194/wes-6-521-2021

Watson, P. A. (2014). Assessing the dependence of surface layer atmospheric stability on measurement height at offshore locations. *Journal of Wind Engineering and Industrial Aerodynamics*, 88-99. doi:https://doi.org/10.1016/j.jweia.2014.06.002